# POLICY-BASED TRAJECTORY CLUSTERING IN OFFLINE REINFORCEMENT LEARNING

## ABSTRACT

We introduce the task of clustering trajectories in offline reinforcement learning (RL) datasets to address the multi-modal nature of offline data. Such datasets often contain trajectories from diverse policies, and treating them as a single distribution can obscure structure and increase distributional shift. We formalize trajectory clustering by linking the KL-divergence of offline trajectory distributions with mixtures of policy-induced distributions.

To solve this, we propose Policy-Guided K-means (PG-Kmeans) and Centroid-Attracted Autoencoder (CAAE). PG-Kmeans iteratively trains behavior cloning policies and assigns trajectories based on generation probabilities, while CAAE adopts a VQ-VAE style objective to guide latent representations toward codebook entries. We prove finite-step convergence of PG-Kmeans and analyze the ambiguity of optimal solutions caused by policy-induced conflicts. Experiments on D4RL and GridWorld show that PG-Kmeans and CAAE partition trajectories into coherent clusters and offer a framework for structuring offline data, with applications in data selection, curriculum learning, and policy transfer.

## 1 INTRODUCTION

Reinforcement learning (RL) has achieved remarkable progress in diverse domains such as robotic control (Tang et al., 2024), autonomous driving (Kiran et al., 2021), and recommendation systems (Lin et al., 2024). However, conventional online RL methods require continuous interaction with the environment, which is often impractical due to high costs and safety concerns, especially in sensitive areas such as healthcare and autonomous driving (Dulac-Arnold et al., 2019). To mitigate these challenges, *offline RL* has emerged as a promising alternative, aiming to learn effective policies from pre-collected datasets without further environment interaction (Levine et al., 2020). The success of offline RL critically depends on how well the available data are organized and utilized.

Offline RL approaches are commonly divided into two categories. (1) *Value-based methods* constrain policies to remain close to the behavior distribution or adopt conservative value estimation to address distributional shift (Fujimoto et al., 2019; Kumar et al., 2019; Yu et al., 2021; Kumar et al., 2020; Kidambi et al., 2021). (2) *Behavior cloning methods* optimize policies via supervised learning on offline trajectories or enforce conservative updates to avoid distributional errors (Chen et al., 2020; Brandfonbrener et al., 2021; Kostrikov et al., 2022). While effective in many cases, both paradigms largely ignore the *structural heterogeneity* of offline datasets—an issue that has increasingly been observed in practice.

Offline datasets are rarely generated by a single policy; instead, they often arise from mixtures of distinct policies with diverse styles. Training on such heterogeneous data can cause mode averaging, interference, and degraded stability. Recent studies provide growing evidence that clustering trajectories is a natural solution, offering several key benefits:

1. **Improved stability and targeted policy learning.** Partitioning trajectories into coherent clusters or modeling them as mixtures improves learning robustness and reduces policy interference (Wang et al., 2024; Osa et al., 2023; Mao et al., 2024). Beyond simply stabilizing training, clustering also enables *cluster-conditioned policy learning*, where agents selectively exploit high-quality clusters or specialize sub-policies to particular behavioral modes, offering finer control over heterogeneous datasets.

2. **Interpretability and behavioral analysis.** Clustering reveals the latent structure of behavior patterns, helping to diagnose suboptimal strategies and summarize complex policies. Prior studies demonstrate that clustering trajectory embeddings (Remman & Lekkas, 2024), applying attention-based abstractions (Bekkemoen & Langseth, 2024), or linking discrete clusters with action attribution (Rishav et al., 2025) can expose distinct behavioral modes. Such analysis not only improves interpretability but also provides a tool for systematically studying how trained agents behave across different contexts.

3. **Data efficiency under limited supervision.** Clustering also supports semi-supervised and sparse-reward scenarios by organizing unlabeled trajectories into meaningful structures. This enables effective learning with limited labels: strong performance can be achieved with as little as 10% reward-labeled data (Zheng et al., 2023), unsupervised skills can be fine-tuned with minimal input (Chebotar et al., 2021), and small sets of preference labels can successfully guide policy optimization when combined with unlabeled data (Kang et al., 2023).

Beyond these points, clustering may also support new paradigms such as *policy-conditioned agents*, where decision-making is explicitly conditioned on cluster identities, or modular policy design in hierarchical RL. While relatively unexplored, these directions highlight the broader potential of clustering as a versatile tool in reinforcement learning.

Despite these promising signs, clustering is typically treated as an auxiliary trick within specific algorithms. The field still lacks a formal problem definition, a theoretical understanding of its properties (e.g., uniqueness of solutions, conflicts between policies), and dedicated algorithms designed for this setting.

This paper establishes policy-based trajectory clustering as a principled research direction for offline RL. Our main contributions are:

1. We formally define the problem of policy-based trajectory clustering, laying the foundation for systematic study (Section 3).

2. We compare policy-based trajectory clustering with traditional clustering methods, and reveal a fundamental connection to the *K-coloring problem*, highlighting unique challenges absent in classical clustering (Section 4).

3. We propose two new algorithms—Policy-Guided K-means (PG-Kmeans) and Centroid-Attracted Autoencoder (CAAE)—that adopt distinct philosophies: the former explicitly maintains central policies per cluster, while the latter uses an encoder–decoder framework for dataset-level clustering (Section 5).

4. We conduct extensive experiments on the D4RL benchmark and custom environments, showing that our methods achieve superior clustering quality under the Normalized Mutual Information (NMI) metric compared to baseline approaches (Section 6).

## 2 RELATED WORK

**Offline Reinforcement Learning.** A central challenge in offline reinforcement learning (RL) is mitigating the distributional shift between the learned policy and the behavior policy from which the dataset was collected. Existing methods can be broadly grouped into three categories. The first constrains the learned policy to remain close to the behavior policy, either explicitly through regularization (Fujimoto et al., 2019; Kumar et al., 2019; Wu et al., 2019), or implicitly via conservative value estimation to prevent reward overestimation (Kumar et al., 2020; Yu et al., 2021). The second emphasizes uncertainty estimation, often employing ensembles to penalize unreliable actions and improve robustness in out-of-distribution regions (Janner et al., 2019; Kidambi et al., 2021). The third includes behavior cloning-based approaches, which either treat behavior cloning as a surrogate for policy learning (Chen et al., 2020), or constrain policy updates to one-step improvements to reduce error accumulation in off-policy evaluation (Brandfonbrener et al., 2021; Kostrikov et al., 2022). In parallel, alternative strategies such as importance sampling and trajectory reweighting aim to directly optimize over trajectory distributions without explicit distributional correction (Zhang et al., 2020; Nachum et al., 2019; Janner et al., 2021). Despite their methodological differences, all of these approaches critically rely on the coverage, quality, and structure of the offline dataset.

**Policy-Based Clustering.** Policy-based clustering has recently emerged as a promising direction for decomposing complex, heterogeneous offline RL datasets into more interpretable and manageable behavior modes. SORL (Mao et al., 2024) and EMPO (Park et al., 2024) adopt expectation-maximization frameworks that alternate between trajectory clustering and policy optimization, enabling the discovery of diverse and high-quality behaviors. Similarly, Wang et al. (2024) propose behavior-aware deep clustering to isolate uni-modal behavioral subsets from multi-behavioral datasets, thereby improving policy stability and performance. Other approaches use probabilistic models for implicit clustering: for instance, Li et al. (2023) employ Gaussian Mixture Models (GMMs) to represent latent behavior policies and derive closed-form policy improvement operators. Diffusion-QL (Wang et al., 2023) leverages expressive diffusion models to capture the multimodal distribution of behavior policies. Collectively, these works highlight the value of policy-level clustering in improving both generalization and robustness in offline RL.

**VAEs in Reinforcement Learning.** Variational Autoencoders (VAEs) (Kingma & Welling, 2013) are widely used in reinforcement learning for unsupervised representation learning, particularly in high-dimensional or visual domains. Applications include model-based RL (Ha & Schmidhuber, 2018), hierarchical skill discovery (Pertsch et al., 2020), and curiosity-driven exploration (Mohamed & Rezende, 2015; Klissarov et al., 2019). A common baseline for trajectory modeling is to embed trajectories into latent spaces using VAEs, followed by clustering with standard algorithms such as K-means. VQ-VAE van den Oord et al. (2018) extends this paradigm by discretizing the latent space via a learned codebook, often improving representation quality and enabling discrete behavior abstraction. However, both continuous VAEs and VQ-VAEs face key limitations: the learned embeddings may not preserve policy-level semantics, and distance-based clustering methods like K-means are poorly aligned with the temporally structured nature of decision-making data (see Table 1). These limitations motivate the design of our CAAE model, which explicitly targets policy-consistent representation learning with clustering objectives in mind.

**Deep Clustering.** Deep clustering methods aim to jointly learn representations and perform clustering in a unified, often end-to-end, framework. A typical approach involves first learning low-dimensional embeddings with neural networks, then applying clustering algorithms such as K-means or GMMs. DEC (Xie et al., 2016) improves cluster assignments via iterative refinement in an autoencoder architecture, while DEPICT (Dizaji et al., 2017) introduces convolutional backbones for image clustering. DAC (Chang et al., 2017) enforces pairwise similarity constraints to enhance representation learning. More recent advancements explore contrastive learning (Li et al., 2020), graph-based methods that exploit structural relationships (Bo et al., 2020), and fully end-to-end clustering models (Ji et al., 2019). These advances provide powerful tools for unsupervised data structuring, many of which can be adapted or extended to sequential decision-making data in RL.

## 3 PROBLEM SETUP: POLICY-BASED TRAJECTORY CLUSTERING

We consider a dataset clustering problem in finite-horizon Markov Decision Process (MDP) offline reinforcement learning (Offline RL), where the objective is to cluster given trajectories based on the policy that generated them.

An MDP is defined as $(\mathcal{S}, \mathcal{A}, P, r, H)$, where $\mathcal{S}$ and $\mathcal{A}$ denote the state and action spaces, respectively. The transition dynamics are given by $P : \mathcal{S} \times \mathcal{A} \to \Delta(\mathcal{S})$, the reward function is $r : \mathcal{S} \times \mathcal{A} \to \mathbb{R}$, and $H$ represents the horizon. A trajectory $\tau$ is defined as $\{s_t, a_t, r_t\}_{t=1}^{H}$ in the full-reward setting or $\{s_t, a_t\}_{t=1}^{H}$ in the reward-free setting, where $s_t \in \mathcal{S}$ and $a_t \in \mathcal{A}$. A deterministic policy $\pi : \mathcal{S} \times [H] \to \mathcal{A}$ maps each state-timestep pair to an action.

We assume the dataset consists of a mixture of $k$ sub-datasets, each collected under a different deterministic behavior policy $\{\pi_i\}_{i=1}^{k}$. Formally,

$$\mathcal{D} = \bigcup_{i=1}^{k} \mathcal{D}_i,$$

where each $\mathcal{D}_i$ is collected by repeating playing $\pi_i$. The objective is to cluster the trajectories according to their source. A alternative objective, considering it as a K-coloring problem, is provided in Appendix A.2.

For evaluation, we denote ground-truth labels and predicted labels by $L$ and $C$, respectively, and use Normalized Mutual Information (NMI) (Strehl & Ghosh, 2002) as the performance metric:

$$\text{NMI}(C, L) = \frac{2 \cdot I(C, L)}{H(C) + H(L)},$$

where $I(C, L)$ is the mutual information between $C$ and $L$, and $H(\cdot)$ denotes Shannon entropy.

# 4    POLICY-BASED TRAJECTORY CLUSTERING VS. TRADITIONAL CLUSTERING

**Nature of Data.**    Traditional deep clustering techniques are typically designed for static data domains such as images, text, or audio. In these settings, individual samples can be embedded into a high-dimensional feature space, and clustering is performed based on similarity in that space. In contrast, offline reinforcement learning datasets comprise sequential state-action pairs generated by latent policies. Here, similarity is not merely a function of geometric proximity in feature space, but also of the underlying generative process, i.e., the decision-making policy.

Due to the stochasticity and high-dimensionality of RL environments, two trajectories that appear similar in state space may in fact arise from fundamentally different policies. As a result, directly applying conventional clustering methods (e.g., K-means or GMM) to raw or embedded state-action sequences often leads to clusters that reflect environmental or task-level structure rather than true policy-level consistency. This misalignment compromises both interpretability and the utility of the clusters for downstream policy learning.

**Ambiguity in Policy-Based Trajectory Clustering: A K-Coloring Perspective**    Unlike classical clustering tasks, policy-based trajectory clustering may not admit a unique solution. To illustrate this, we draw a reduction to the well-known *K-coloring problem*. Specifically, we construct a graph where each node represents a trajectory, and an edge connects two nodes if the corresponding trajectories exhibit conflicting decision behavior—i.e., they cannot have been generated by the same stationary policy. In this formulation, a valid K-coloring corresponds to a feasible partition of trajectories into policy-consistent clusters.

We further provide an inverse reduction (see Appendix A.2) and show that the policy-based trajectory clustering problem is NP-complete. This theoretical result highlights a core distinction from standard clustering: the existence of multiple, equally valid clusterings. While this introduces inherent ambiguity, empirical evidence suggests that stable and semantically meaningful solutions can still be obtained under realistic assumptions on the data distribution (see Appendix A.3).

# 5    METHODS

To address the problem of policy-based trajectory clustering, we consider two general approaches.

First, we propose Policy-Guided KMeans (PG-Kmeans), which explicitly maintains $k$ distinct clusters along with their corresponding policy centroids for clustering. This method operates in the space of behaviors by aligning trajectories with representative policies.

Alternatively, in the direction of representation learning, we introduce the Centroid-Attracted Autoencoder (CAAE), which trains an encoder to map trajectories into a low-dimensional latent space. Clustering is then performed in this latent space based on the distance between the embedded representation and a set of learnable centroids.

## 5.1    POLICY-GUIDED KMEANS

The algorithmic foundation of PG-Kmeans is similar to that of the standard K-means clustering algorithm. In K-means-like algorithms, the clustering objective is typically defined as the sum of distances between data points and their respective cluster centers, which is optimized using the Expectation-Maximization (EM) method.

We derive our clustering objective from an information-theoretic perspective, minimizing the *Kullback-Leibler (KL) divergence* between the empirical data distribution $P(\tau)$ and the mixture of policy-

induced distributions $\hat{P}(\tau)$, where

$$\hat{P}(\tau) = \sum_{j=1}^{k} w_{i,j}\mathbb{P}(\tau|\theta_j).$$

Here, $\mathbf{W} = \{w_{i,j}\}_{(i,j)=(1,1)}^{(N,k)}$ represents cluster assignments, $N$ is the number of data points, and $k$ is the number of clusters. $w_{i,j}$ is the indicator of trajectory $\tau_i$ being assigned to the cluster $j$. The KL divergence quantifies the discrepancy between these distributions:

$$D_{\mathrm{KL}}(P(\tau)\|\hat{P}(\tau)) = \mathbb{E}_{\tau \sim P}\left[\log \frac{P(\tau)}{\hat{P}(\tau)}\right]. \tag{1}$$

Minimizing this divergence ensures that the learned policies $\{\pi_j\}_{j=1}^{k}$ effectively approximate the empirical data distribution. Since $P(\tau)$ is independent of the optimization variables $\theta_j$ and $w_{i,j}$, minimizing KL divergence is equivalent to maximizing:

$$\mathbb{E}_{\tau \sim P}\left[\log \hat{P}(\tau)\right] \approx \sum_{i=1}^{N} \log \hat{P}(\tau_i) = \sum_{i=1}^{N}\sum_{j=1}^{k} w_{i,j} \log \mathbb{P}(\tau_i|\theta_j). \tag{2}$$

Here we denote $\tau_i$ as the sequence $\{(s_{i,h}, a_{i,h})\}_{h=1}^{H}$. Noting that the assignment weights are binary (0 or 1) and that the environment dynamic is independent of $\theta_j$ and $w_{i,j}$, we arrive at the final objective function:

$$\underset{\theta, \mathbf{W}}{\text{maximize}}\, J(W, \theta) = \sum_{i=1}^{N}\sum_{j=1}^{k} w_{i,j} \sum_{h=1}^{H} \log \mathbb{P}(a_{i,h}|\theta_j, s_{i,h}), \tag{3}$$

$$\text{s.t.} \quad w_{i,j} \in \{0,1\},\, \forall(i,j) \in [N] \times [K],$$

$$\sum_{j=1}^{k} w_{i,j} = 1,\, \forall i \in [N].$$

This objective function formulates an optimal distribution matching problem, where we seek to approximate the empirical data distribution using a mixture of policy-induced distributions. The cluster assignments $w_{i,j}$ ensure that each data point is assigned to the most suitable policy, while the policy parameters $\theta_j$ are optimized to maximize the likelihood of the assigned trajectories.

### 5.1.1 MAIN ALGORITHM

To solve the clustering objective, we derive an iterative algorithm based on the Expectation-Maximization (EM) framework. See Algorithm 1 for the pseudocode. Similar to K-means, the algorithm alternates between an E-step, where data points are assigned to the most probable generating cluster, and an M-step, where each cluster center is updated to maximize the likelihood of generating its assigned trajectories. This process iterates until convergence, after which cluster centers may be merged to further optimize the objective function $J(W, \theta)$.

The idea of explicitly maintaining $k$ central policies and clustering trajectories based on the likelihood that each central policy generates a target trajectory has also been explored in SORL Mao et al. (2024). However, it is important to note that SORL employs a continuous assignment matrix $\mathbf{W}$, making it more akin to an EM-style method than to KMeans. As a result, SORL is more prone to subpopulation homogenization, a phenomenon also observed in our experiments (see Table 1).

Moreover, due to the use of soft assignments via $\mathbf{W}$, SORL requires training each policy network over the entire dataset. In contrast, PG-Kmeans leverages the classification results from the previous round as a prior approximation, and assigns each data point to a separate network, thereby significantly reducing the overall training cost.

---

**Algorithm 1** Policy-Guided Kmeans (PG-Kmeans)

---

1: **Input:** Dataset $\mathcal{D} = \{\tau_1, \ldots, \tau_N\}$, number of clusters $k$, ground truth $k^*$ (optional), maximum iterations $T$.
2: **Initialize** cluster assignments $W$ randomly.
3: **Initialize** $k$ policies $\{\pi_1, \pi_2, \ldots, \pi_k\}$ using behavior cloning (BC) trained on the respective clusters.
4: **for** $t = 1$ to $T$ **do**
5:     **M-step:** Update each policy $\pi_j$ by training a behavior cloning model on the trajectories assigned to cluster $j$.
6:     **E-step:** Assign each trajectory $\tau_i \in \mathcal{D}$ to the cluster $j = \arg\max_{j'} \mathbb{P}(\tau_i|\theta_{j'})$, i.e., $w_i = e_j$, where $e_j$ is a one-hot vector.
7:     **Check Convergence:** If cluster assignments $W$ remain unchanged, terminate.
8: **end for**
9: If $k^*$ is given, run Algorithm 4 to merge clusters.
10: **Output:** Final cluster assignments $W$ and policies $\{\pi_1, \pi_2, \ldots, \pi_k\}$.

---

### 5.1.2 THEORETICAL ANALYSIS

Theoretically, we establish the convergence of PG-Kmeans following a similar argument as K-means (Bottou & Bengio, 1995). Before convergence, the loss function strictly decreases after each iteration, ensuring that no identical grouping pattern occurs during the training process. Since the number of possible grouping patterns for $N$ data points is finite, PG-Kmeans is guaranteed to converge within a finite number of iterations.

**Theorem 5.1 (Finite-Step Convergence of PG-Kmeans)** *Given a dataset with $N$ trajectories and $k$ clusters, the PG-Kmeans algorithm is guaranteed to converge within a finite number of iterations.*

Rigorous proof deferred to Appendix A.1. We note that in the worst case, the number of iterations required can be as high as $O\left(k^N\right)$. However, in our experimental setup, the algorithm consistently converged within 20 iterations.

### 5.1.3 IMPLEMENTATION DETAILS OF PG-KMEANS

In practice, the vanilla version of PG-Kmeans exhibited considerable instability during training. To address this, we adopted a best-of-$N$ selection strategy based on training loss to improve robustness.

In addition, we employed an overparameterization-and-merging approach to handle scenarios where the true number of clusters $k$ is unknown. Empirically, we found that initializing PG-Kmeans with a slightly overestimated number of hypothetical cluster centers leads to substantial performance improvements (see Section 6.2). For further details, please refer to Appendix B.4.

## 5.2 CENTROID-ATTRACTED AUTOENCODER (CAAE)

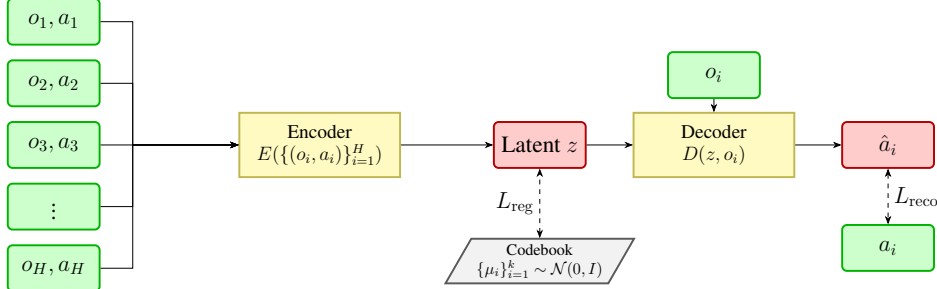

Figure 1: A schematic of our CAAE. The full trajectory $\{(o_i, a_i)\}_{i=1}^H$ is first encoded into a compact latent vector $z$. A learnable Gaussian codebook $\{\mu_i\}$ imposes a regulation loss to keep $z$ near the prior, while the decoder conditions on both $z$ and a single observation $o_i$ to reconstruct the action $\hat{a}_i$.

In addition to the explicit trajectory clustering approach employed by PG-Kmeans, we introduce an alternative representation learning-based method, the Centroid-Attracted Autoencoder (CAAE). In CAAE, each input trajectory $\tau$ is first encoded into a latent representation $z = \text{encoder}(\tau)$, which is then combined with the observation sequence and passed through a decoder to reconstruct the trajectory $\tilde{\tau} \sim \text{decoder}(z, o_{1:H})$.

To regularize the latent space, we impose a constraint motivated by the assumption that each latent variable follows a Gaussian distribution, i.e., $z_i \sim \mathcal{N}(\mu_i, I)$, where $\mu_i$ is selected from a learnable codebook $\{\mu_j\}_{j=1}^{k}$. This assumption leads to a penalty term that encourages each $z_i$ to be close to one of the centroids in the codebook, thus promoting structured and interpretable embeddings.

Formally, the CAAE objective is defined as:

$$L(\phi, \theta, \mu) = \sum_{i=1}^{N} \left( -\sum_{h=1}^{H} \log \mathbb{P}(a_{i,h} \mid \theta, z_i, s_{i,h}) + \min_{j} \|\mu_j - z_i\|^2 \right) - \frac{1}{k^2} \sum_{i,j} \min\{1, \|\mu_i - \mu_j\|_2^2\},$$

(4)

where $z_i = \text{encoder}(\tau_i, \phi)$, $\theta$ and $\phi$ represents the parameters of encoder and decoder respectively, and the last term is a regularization term. Implementation details can be found in Appendix B.5 and detailed discussion about the difference between CAAE and VQ-VAE(van den Oord et al., 2018) can be found in Appendix E.

## 5.3 Comparison Between PG-Kmeans and CAAE

As one of the first systematic investigations into policy-based trajectory clustering in reinforcement learning, our work introduces and contrasts two complementary approaches—PG-Kmeans and CAAE—each reflecting a distinct design philosophy and offering unique advantages. The differences between these methods can be primarily characterized along two key dimensions:

First, PG-Kmeans assigns each cluster a separate policy network, whereas CAAE employs a shared decoder modulated by latent inputs from a unified encoder. This architectural distinction leads PG-Kmeans to produce sharper, more distinct cluster boundaries, enabling precise policy segmentation. However, this comes at the cost of higher computational complexity and increased sensitivity during training. In contrast, CAAE benefits from parameter sharing, resulting in greater computational efficiency and empirically more stable optimization.

Second, PG-Kmeans performs clustering based on single-step action likelihoods, making it effective at capturing fine-grained distinctions within mixed-intent or concatenated trajectories. CAAE, by comparison, encodes entire trajectories and performs clustering in the latent space of these holistic embeddings. This makes it more adept at modeling global behavioral structure, but less responsive to localized transitions or mode-switching within a single trajectory.

By proposing both PG-Kmeans and CAAE, we offer a dual-perspective framework for policy-based trajectory clustering—one that balances precision and generalization, and accommodates a diverse set of practical scenarios and modeling requirements in offline RL.

## 6 Experiments

We evaluate PG-Kmeans and CAAE in both continuous and discrete action spaces and compare it against several traditional clustering methods.

### 6.1 Environments and baselines

**Gridworld.** We designed three discrete environments and a continous environment with corresponding policies: Takeball, Diagonal and Extra as discrete environment and Pathfollowing as continous environment (Figure 2). In the Takeball environment, the agent selects one of four different balls on the map before navigating to the bottom-right corner as the destination. In the Diagonal environment, the agent is randomly initialized in the top-left corner and must reach the bottom-right corner. In the Extra environment, the agent need to move from a start grid to a terminal grid in a randomly generated map, with two specially marked grids. In Pathfollowing, the agent need to control a ball to

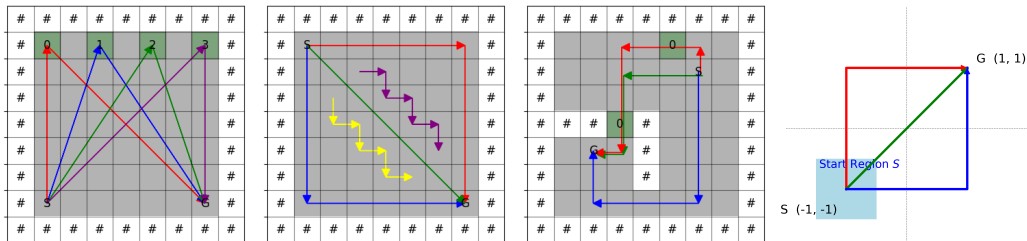

Figure 2: **Takeball (left 1), Diagonal (left 2), Extra (left 3), Pathfollowing (left 4).** In Takeball, the four ground-truth policies correspond to tendencies to collect each of the four balls, respectively. In Diagonal, Expert 1 (red) prefers moving right first; Expert 2 (blue) prefers moving down first; Expert 3 (green) follows the diagonal; Experts 4 (purple) and 5 (yellow) follow zigzag paths. In Extra, Expert 1 (red) always visits the two special grids; Expert 2 (blue) never visits them; Expert 3 (green) ignores the special grids. In Pathfollowing, three experts try to move corresponding routes first and then follows the route to the terminal point, the special grids are labeled by 0.

Table 1: Normalized Mutual Information (NMI) scores for all methods on D4RL and Gridworld environments, The values in the table are presented as mean $\pm$ std. VAE and Return are represent algorithm VAE+Kmeans and Return+Kmeans. Each score is calculated over at least 10 random seeds. Our algorithms, PG-Kmeans and CAAE, demonstrate consistently high clustering accuracy across the majority of datasets. In contrast, baseline methods such as DEC, SORL, and Return+Kmeans perform well only on a limited subset of datasets.

| Task | PG-Kmeans | CAAE | VAE | VQ-VAE | Return | DEC | SORL |
|---|---|---|---|---|---|---|---|
| Halfcheetah | **0.99** $\pm$ 0.00 | **0.99** $\pm$ 0.00 | 0.00 $\pm$ 0.00 | 0.96 $\pm$ 0.01 | 0.97 $\pm$ 0.00 | 0.95 $\pm$ 0.01 | 0.12 $\pm$ 0.33 |
| Ant | 0.92 $\pm$ 0.00 | **0.96** $\pm$ 0.01 | 0.01 $\pm$ 0.01 | 0.68 $\pm$ 0.11 | 0.05 $\pm$ 0.00 | 0.39 $\pm$ 0.17 | 0.00 $\pm$ 0.00 |
| Walker2d | **0.94** $\pm$ 0.01 | 0.88 $\pm$ 0.10 | 0.00 $\pm$ 0.00 | 0.59 $\pm$ 0.18 | 0.23 $\pm$ 0.00 | 0.77 $\pm$ 0.12 | 0.08 $\pm$ 0.24 |
| Hopper | **0.99** $\pm$ 0.00 | **0.99** $\pm$ 0.01 | 0.00 $\pm$ 0.00 | 0.80 $\pm$ 0.20 | 0.86 $\pm$ 0.00 | 0.00 $\pm$ 0.00 | 0.84 $\pm$ 0.26 |
| Diagonal | **0.92** $\pm$ 0.00 | 0.87 $\pm$ 0.05 | 0.10 $\pm$ 0.02 | 0.16 $\pm$ 0.03 | N/A | 0.28 $\pm$ 0.02 | 0.18 $\pm$ 0.05 |
| Takeball | **1.00** $\pm$ 0.00 | **1.00** $\pm$ 0.00 | 0.00 $\pm$ 0.00 | 0.57 $\pm$ 0.15 | N/A | 0.05 $\pm$ 0.10 | 0.54 $\pm$ 0.13 |
| Pathfollowing | 0.12 $\pm$ 0.11 | 0.15 $\pm$ 0.02 | 0.03 $\pm$ 0.05 | 0.09 $\pm$ 0.04 | N/A | **0.26** $\pm$ 0.10 | 0.14 $\pm$ 0.05 |
| Extra | 0.02 $\pm$ 0.01 | **0.43** $\pm$ 0.22 | 0.04 $\pm$ 0.05 | 0.15 $\pm$ 0.16 | N/A | 0.28 $\pm$ 0.36 | 0.00 $\pm$ 0.00 |

be close to a specific point. For each environment, we designed 3-5 different strategies and collected a balanced dataset with them.

To introduce stochasticity, we applied a 0.3 probability of random dynamics at each step in discrete environments, and added Gaussian noise to each move in Pathfollowing. The initial position was sampled uniformly from $[-1.5, -0.5]^2$ in Pathfollowing. The reward function is trivially set to a constant 0. See Appendix B.1 for details.

**Gym Environments.** For continuous tasks, we also used the widely adopted D4RL dataset (Fu et al., 2020). Specifically, we selected the medium-expert datasets from four Gym environments, as they align well with the definition of datasets composed of multiple deterministic policies.

**Baseline Methods** To the best of our knowledge, no algorithm has been developed specifically for policy-based clustering prior to this work. Therefore, in our experiments we compare against a selection of deep clustering methods and functionally analogous approaches. Specifically, we use Deep Embedded Clustering(Xie et al., 2016), SORL(Mao et al., 2024), Return + Kmeans, VQ-VAE(van den Oord et al., 2018) and VAE + Kmeans as our baselines.

## 6.2 RESULTS AND DISCUSSION

As shown in Table 1, both PG-Kmeans and CAAE achieve consistently high NMI scores across most datasets. PG-Kmeans outperforms SORL in nearly all configurations, highlighting the advantage of

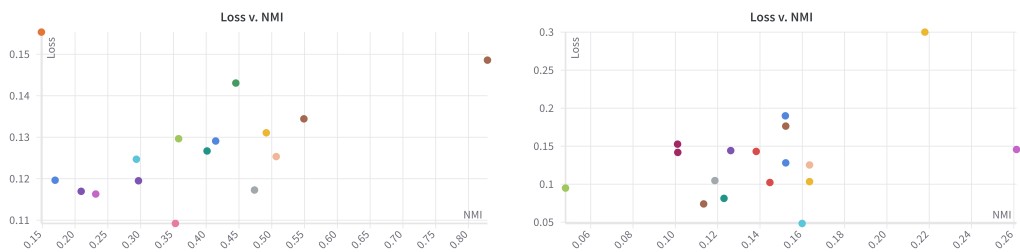

Figure 3: NMI vs. loss in Extra(Left) and Pathfollowing(Right) with CAAE. The figure shows no significant correlation between NMI and loss, indicating overfitting.

explicitly maintaining central policies. Conventional representation learning-based approaches, such as DEC or VAE with K-means, generally fail to provide meaningful clustering.

A noteworthy baseline is VQ-VAE, which already surpasses DEC and in many environments even outperforms SORL by leveraging discrete latent codes. Nevertheless, CAAE achieves a clear margin over VQ-VAE. By learning continuous latent representations rather than relying on codebook projection, CAAE avoids quantization errors and early-stage instability caused by the Straight-Through Estimator, leading to faster convergence and more stable clusters. As illustrated in Figure 7, a t-SNE visualization of the TakeBall environment shows that CAAE produces compact clusters aligned with policy semantics, while VQ-VAE clusters remain fragmented.

We also note that in certain environments, even the best algorithms fail to achieve an NMI close to 1.0. This gap is partly due to optimization but also reflects inherent indistinguishability in the data. For example, in the Diagonal environment, Expert 3 and Expert 4 exhibit nearly identical behavior near the diagonal, making separation difficult. Similarly, in Extra, Experts 2 and 3 sometimes follow almost identical paths on specific maps.

Another important challenge is overfitting (Figure 3). When the conflict rate of $(s, a)$ pairs between two policies is low, merging them incurs only minor loss increase. For instance, in PathFollowing, policies conflict mainly in the start region, allowing a single network to approximate all policies with relatively low loss—even if it fails to distinguish them accurately.

Finally, ablation studies on the choice of cluster number $k$ and the role of regularization (Appendix C.2) confirm that regularization is crucial, and mild overparameterization can further improve performance.

## 7 SUMMARY AND LIMITATIONS

In this paper, we formalized the problem of policy-based trajectory clustering in offline reinforcement learning and proved that it is NP-complete. We further analyzed how it differs from conventional clustering tasks. To address this challenge, we introduced two methods—PG-Kmeans and CAAE—and evaluated them on carefully curated datasets. Experimental results demonstrate their clear advantages over existing baselines.

A key limitation of this study is the relatively small scale of the experiments. The proposed methods have not yet been tested on large-scale datasets or in complex, real-world environments. Moreover, the algorithms currently lack theoretical convergence guarantees, and the uniqueness of clustering solutions remains an open question. Future work may involve a more rigorous problem formulation, improved strategies to mitigate overfitting, and a comprehensive theoretical analysis of the proposed approaches.

## ACKNOWLEDGMENTS AND DISCLOSURE OF FUNDING

SSD acknowledges the support of NSF DMS 2134106, NSF CCF 2212261, NSF IIS 2143493, NSF IIS 2229881, Alfred P. Sloan Research Fellowship, and Schmidt Sciences AI 2050 Fellowship.

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

CONTENTS

## A APPENDIX: MATHMATICAL DISCUSSION

### A.1 PROOF OF THEOREM 5.1

**Theorem A.1 (Finite convergence of Algorithm 1)** *Algorithm 1 is guaranteed to converge in* $O(k^N)$ *iterations.*

**Proof A.1** *We denote the original assignment matrix* $W$ *as* $W^0$, *policy parameters as* $\theta^0$. *After iteration* $t$, *the assignment matrix and parameters become* $W^t$ *and* $\theta^t$. *Assume that the policy converges at after* $T \in \mathbb{Z}^+ \cup \{\infty\}$ *iterations. From the definition we see that* $\forall t = 1, 2, \cdots T$,

$$\max_{\theta} J(W^{t-1}, \theta) = J(W^{t-1}, \theta^t) \tag{5}$$

$$< \max_{W} J(W, \theta^t) \tag{6}$$

$$= J(W^t, \theta^t) \tag{7}$$

$$\leq \max_{\theta} J(W^t, \theta). \tag{8}$$

*Therefore* $\{\max_{\theta} J(W^{t-1}, \theta)\}_{t=1}^T$ *is a strictly increasing sequence. Because there are at most* $k^N$ *different valid values for* $W$, *the sequence length* $T$ *is no larger than* $k^N$, *which concludes the proof.*

### A.2 POLICY-BASED TRAJECTORY CLUSTER AND K-COLORING

One fundamental reason why K-means clustering cannot be directly applied to policy-based trajectory clustering is the absence of a well-defined distance metric in trajectory space. A natural idea is to define the "distance" between two trajectories based on their compatibility, i.e., whether they could have been generated by the same policy:

$$d(\tau_1, \tau_2) = \begin{cases} 1, & \exists (s, a) \in \tau_1, (s, a') \in \tau_2, a \neq a' \\ 0, & \text{otherwise} \end{cases} \tag{9}$$

However, this definition does not satisfy the triangle inequality, i.e.,

$$|d(x, y)| \leq |d(x, z)| + |d(z, y)| \tag{10}$$

which means it is not a proper metric and provides limited mathematical utility.

For example, consider the trajectories $x = [(s_1, a_1)]$, $y = [(s_1, a_2)]$, and $z = [(s_2, a_1)]$. In this case, we observe that:

$$|d(x, y)| = 1 \nleq |d(x, z)| + |d(z, y)| = 0. \tag{11}$$

We also note that, in the absence of additional assumptions, Equation (9) encapsulates all the reliable information available in the dataset. That is, any clustering scheme $W$ that satisfies the condition that the total intra-cluster distance is zero provides a valid solution:

$$D(W) = \sum_{k=1}^{K} \sum_{i=1}^{N} \sum_{j=1}^{N} w_{i,k} w_{j,k} d(\tau_i, \tau_j) = 0. \tag{12}$$

This formulation precisely aligns with the definition of the K-coloring problem.

Furthermore, we can construct a simple proof for the following theorem:

**Theorem A.2 (Reduction from K-coloring to Policy Clustering)** *For any K-coloring problem with $N$ nodes and a maximum degree of $d$, there exists a Markov Decision Process (MDP) with $|\mathcal{S}| \geq 2d$ and $|H| \geq d$, along with a corresponding dataset, such that the dataset can be generated by $K$ distinct policies, and its valid clustering corresponds to a solution of the original K-coloring problem.*

**Proof A.2** *We can construct a dataset by following steps:*

---

**Algorithm 2** Reduction from K-coloring to policy clustering

---

1: **Input:** a graph $G = (V, E), s.t. \max(\text{degree}(v)) = d$
2: **Initialize** Initialize a list $t_i$ for each node $v_i$.
3: **for** $i = 1$ to $|V|$ **do**
4:     **for** $j = i + 1$ to $|V|$ **do**
5:         If $(v_i, v_j) \in E$, find the smallest integer $l$ such that $s_l \notin t_i \cup t_j$
6:         Append $t_i$ with $(s_l, a_1)$, $t_j$ with $(s_l, a_2)$
7:     **end for**
8: **end for**
9: Pad $\{t_i\}_i$ to length $H$ and concatenate them to get trajectories $\{\tau_i\}_i$.
10: **Output:** Dataset for clustering $\{\tau_i\}_{i=1}^{|V|}$.

---

*Because $|t_i \cup t_j| \leq \text{degree}(v_i) + \text{degree}(v_j) \leq 2d$, $l$ would never take value over $2d$. And $\max_i(|t_i|) \leq \max(\text{degree}(v)) = d$, so horizon $H > d$ is enough.*

This result implies that the general policy-guided clustering problem is NP-complete, for we have known that 3-coloring with $d \geq 2$ is NP-complete.

## A.3 POLICY AMBIGUITY

The above reduction from k-coloring to policy-based clustering demonstrates that policy-based clustering may have multiple solutions. However, this is not necessarily unacceptable. As long as the center policies of the clusters remain consistent, different clustering solutions merely arise due to the fact that these policies exhibit identical expressions in certain instances. The more critical issue is that not only can the trajectory clustering solutions be non-unique, but the center policies themselves may also constitute entirely different constructure. Consider the following example:

Let us examine a contextual bandit setting with only two states and two actions. This environment allows for exactly four distinct deterministic policies and four possible trajectories:

- **Policy 1:** $(s_1, a_1), (s_2, a_1)$
- **Policy 2:** $(s_1, a_1), (s_2, a_2)$
- **Policy 3:** $(s_1, a_2), (s_2, a_1)$
- **Policy 4:** $(s_1, a_2), (s_2, a_2)$

Using either Policy 1 and Policy 4, or Policy 2 and Policy 3, we can generate all four possible trajectories. Thus, for any dataset collected from this contextual bandit, there will always exist at least two valid clustering solutions. Furthermore, when the occurrence probabilities of $s_1$ and $s_2$ are equal, these two clustering solutions are completely irrelevant with each other.

The experimental results demonstrate that the non-uniqueness of policy combinations is not merely a theoretical possibility but frequently manifests in practical datasets. In the deprecated environment analogous to Takeball, we observe a modified scenario where four balls are randomly permuted across four fixed positions, while the expert policy $\pi_i$ still maintains the strategy of collecting the $i$-th numbered ball and returning.

Notably, the neural network rapidly learns a distinct policy categorization approach: instead of differentiating policies by ball numbering, it develops four position-specific policies that collect balls based on their spatial locations. Although both policy combinations generate identical trajectory distributions within the environment, the position-based strategy exhibits significantly simpler implementation requirements. Each agent in this paradigm only needs to learn navigation to a fixed location, avoiding the more complex task of simultaneously identifying specific ball numbers and selecting among four potential destinations.

Through empirical analysis, we find that random initialization almost invariably leads the network to discover the position-based discrimination method. In contrast, achieving number-based trajectory differentiation requires initializations remarkably close to the target clustering configuration (specifically, Normalized Mutual Index (NMI) > 0.6 in our experiments).

This fundamental non-uniqueness in policy combinations constitutes a critical challenge that directly impacts problem solvability, as it introduces substantial ambiguity in policy identification and may lead to suboptimal solutions that fail to capture the intended behavioral semantics.

## B  APPENDIX: IMPLEMENTATION DETAILS

### B.1  DATASET SETTINGS

**D4RL**   We directly use the medium-expert datasets provided by D4RL for four Gym environments. Each dataset consists of 1,000,000 timesteps. The original datasets are not segmented into episodes nor labeled with their generating policies. Therefore, we divide the dataset into episodes based on the 'Done' signal and assign labels accordingly.

Since the first half of the dataset corresponds to medium-level data and the latter half to expert-level data, we partition the dataset by maximizing the average return of the first and second halves. However, as it is possible that some of the initial expert episodes achieve relatively low returns, this method may introduce an error of up to five episodes (approximately 5,000 timesteps). Nonetheless, this minor misclassification has a negligible impact on the final NMI and does not affect the quantitative conclusions presented earlier.

**Gridworld**   The Diagonal, Takeball and Extra environments are implemented in JAX. To align with the MDP framework, we use a fixed 9×9 grid map. The action space consists of five discrete actions: movement in four directions and a standby action. The maximum episode length is set to 40 timesteps, and an episode terminates immediately when the agent reaches the goal position. Additionally, to simulate stochastic dynamics, each step has a 0.3 probability of ignoring the action input and taking a random action.

The observation space contains the full state information. In Diagonal, the observation is represented as a 9×9×3 matrix obtained by stacking the wall map, agent map, and goal map. In Takeball, the observation further includes four additional one-hot matrices to encode the positions of four balls, resulting in a 9×9×4 representation. In Extra, the observation further includes two additional one-hot matrices to encode the positions of two special grid, resulting in a 9×9×2 representation.

For Diagonal and Takeball environments, all nine expert policies are rule-based, relying solely on the current state without considering historical information. They are implemented in JAX. For each expert, we collect 20,000 trajectories for evaluation. As a result, the Diagonal dataset consists of a total of 100,000 trajectories, while the Takeball dataset contains 80,000 trajectories. And for Extra environment. We train three policies using PPO algorithm by setting positive/zero/negative rewards for the agent to reach the special grid. Other details are the same as Diagonal and Takeball. The Extra dataset consists of a total of 60,000 trajectories.

### B.2  DATASET COLLECTION METHODS IN GRIDWORLD

**Diagonal**   There are five different rule-based policies:

1. Always move to the right, until the wall is reached.
2. Always move to the down, until the wall is reached.

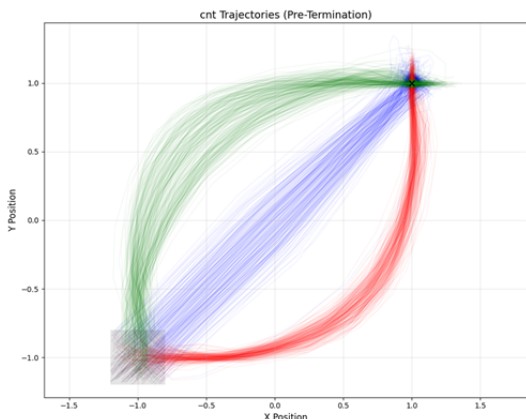

Figure 4: Each color represents one expert policy in PathFollowing.

3. Move to the right iff at the left-down half of the gridworld.

4. Move to the right iff at the black grid, if we seem the gridworld as a chess board.

5. Move to the right iff at the white grid.

**Takeball**  There are four different rule-based policies, i-th policy will pick the i-th ball first.

**Extra**  We use PPO algorithm to train three policies by giving positive/zero/negative rewards for the agent to reach the special grid. The rewards are set to +10, 0 or -10, respectively. When the agent reach the goal, the reward is set to +10. The agent will get a penalty of -0.3 for each step and each unit of distance to the goal.

**PathFollowing**  We use PPO algorithm to train three policies by giving extra penalty for the agent when it off path. For each step, the reward is negative distance to the goal add the following penalty for each policy:

1. No extra penalty.
2. 5 times of distance to the polyline $(-1,-1),(-1,1),(1,1)$.
3. 5 times of distance to the polyline $(-1,-1),(1,-1),(1,1)$.

These policies are corresponding to: No preference, prefer to go up first and prefer to go right first.Figure 4 illustrates the trajectories associated with different datasets.

B.3  BASELINE METHODS

**Deep Embedded Clustering (DEC).**  Deep Embedded Clustering (DEC) Xie et al. (2016) is a widely used deep clustering method that integrates representation learning with clustering optimization. It first pretrains an autoencoder to map high-dimensional data into a low-dimensional latent space. Clustering is then performed in this learned space by iteratively refining cluster assignments using a self-training objective. Specifically, DEC employs a Student's t-distribution to measure the similarity between data points and cluster centers and minimizes a Kullback-Leibler (KL) divergence loss to refine embeddings for more compact and well-separated clusters.

In this work, we adapt DEC for policy clustering by modifying its encoding process to focus on capturing policy-related information rather than trajectory-level dynamics. The details are attached in Appendix B.5.

**SORL**  Stylized Offline Reinforcement Learning(SORL) Mao et al. (2024) is a offline reinforcement learning algorithm that designed to learn diverse policies with varied styles. The first step of the algorithm is soft clustering, that is assigning weight to each trajectories on clusters. In this paper, we

only adopt the clustering component of SORL, without performing the subsequent reinforcement learning stage. Specifically, we execute lines 1–11 of Algorithm 1 from Mao et al. (2024) and then cluster trajectories according to the posterior distribution $\hat{p}(z \mid \tau)$ as defined in the original paper.

**VAE, VQ-VAE**  VAE learns a continuous latent representation by mapping inputs to a Gaussian distribution and reconstructing them through a probabilistic decoder. This structure encourages smooth latent interpolation and provides a flexible foundation for unsupervised representation learning. Our VAE baseline is constructed by first training a standard VAE model, then applying the K-means algorithm directly to the latent variables obtained from encoding each trajectory.

VQ-VAE incorporates a discrete codebook to represent latent variables, replacing the continuous latent distribution used in CAAE. This quantization mechanism constrains representations to a finite set of embeddings, helping stabilize training and promoting structured latent reuse. Beyond this latent design, the architecture and training configuration remain fully aligned with CAAE for a fair comparison.

To ensure a fair comparison, our VQ-VAE and VAE implementation adopts the same encoder, decoder, reconstruction loss, and all hyperparameters, if applicable, as CAAE. The primary difference between VQ-VAE and CAAE lies in the design and usage of the latent variables. A more detailed analysis of their distinctions can be found in Appendix E.

**Returns + Kmeans**  The *returns+K-means* baseline performs clustering solely based on the return of each trajectory using the K-means algorithm. Consequently, it cannot be applied in scenarios where return information is unavailable. As shown in our results, this method performs poorly in environments with high stochasticity, such as *Ant* and *Walker2d*.

### B.4    DETAILS OF PG-KMEANS IMPLEMENTATION

### B.4.1    BEST-OF-N PG-KMEANS

Similar to K-means, PG-Kmeans is highly sensitive to initialization. Moreover, since PG-Kmeans optimizes a neural network for each cluster, small clusters are particularly prone to severe overfitting and non-convex optimization issues, which can ultimately lead to their disappearance.

This instability makes PG-Kmeans less robust during training. To mitigate this issue, we propose the Best-of-N technique. As shown in Algorithm 3, we use the final objective function $J(W, \theta)$ as an internal metric to evaluate the quality of different runs in the absence of ground truth. The best clustering result is then selected for output.

---

**Algorithm 3** Best-of-N PG-Kmeans

---

1: **Input:** Input for PG-Kmeans and number of runs $N$.
2: **for** $t = 1$ to $N$ **do**
3:    Run PG-Kmeans (Algorithm 1) and obtain the corresponding output and $J(W, \theta)$.
4: **end for**
5: **Output:** Result from the run with the highest $J(W, \theta)$.

---

### B.4.2    OVER-PARAMETERIZATION AND MERGING

To further improve optimization performance, we introduce the over-parameterization and merging technique. The algorithm faces two primary challenges: (1) center policies often overfit to short and low-density trajectories, causing them to become trapped in incorrect clusters, and (2) with suboptimal initialization, different clusters may be mistakenly merged during clustering. Over-parameterization mitigates these issues by initializing the cluster count $k$ larger than the true number of clusters $k^*$, enhancing clustering robustness. However, this approach introduces a new challenge: data points from the same cluster may become dispersed, reducing clustering quality. The merging step addresses this issue by consolidating similar clusters, counteracting the adverse effects of a large $k$ while preserving clustering coherence.

For detailed experimental results, see Section 6.2.

---

**Algorithm 4** Merge Clusters

---

1: **Input:** Datasets and center policies $\{(\mathcal{D}_i, \pi_i)\}_{i=1}^k$, target number of clusters $k^*$.
2: **for** $i = 1$ to $k - k^*$ **do**
3:    Find indices $i, j = \arg\min_{i \neq j} \sum_{\tau \in \mathcal{D}_j} \log \mathbb{P}(\tau \mid \theta_i)$.
4:    Merge dataset $j$ into dataset $i$, discard policy $\pi_j$. Renumber datasets and policies from 1 to $k - i$.
5: **end for**
6: **Output:** Datasets and center policies $\{(\mathcal{D}_i, \pi_i)\}_{i=1}^{k^*}$.

---

## B.5 DETAILS OF REPRESENTATION LEARNING-BASED ALGORITHM IMPLEMENTATION

In practice, we found that the most important information to distinguish different strategies for a trajectory are given by a contiunous sub-trajectory. Inspired by this observation, we design the encoder model as follows: use a GRU network to encode every prefix and use attention mechanism to get the weighted sum of the outputs of GRU. That is, if the output of GRU is $Y = (y_1, \ldots, y_H)$, we will train three matrix $Q, K, V$, then the output of attention is:

$$a = \text{softmax}(Q(KY)^\top)(VY)$$

During decoding, we condition on the observation at each timestep along with the embedding to generate the action distribution, using the negative log-likelihood of the true action under this distribution as the reconstruction loss. Under this formulation, the encoder is explicitly designed to capture only policy information, i.e., the action generation mechanism.

To highlight the advantages of CAAE over other representation learning-based algorithms such as DEC and VAE, we employed the same Encoder and Decoder structures used in CAAE for DEC and VAE.

## B.6 NETWORK ARCHITECTURES AND TRAINING DETAILS

All networks are trained using the Adam optimizer with a learning rate of $1 \times 10^{-3}$. The parameter $\alpha$ of CAAE is set to 1. For the D4RL environments, observations are pre-normalized using statistics from the training set, while no normalization is applied to GridWorld inputs.

**Policy Networks.** For continuous environments, policies use a `MultivariateNormalDiag` distribution, where two fully connected (FC) layers process extracted features to produce `action_logits` and `action_std`. For discrete environments, policies use a `Categorical` distribution, with a single FC layer mapping extracted features to `action_logits`.

**Feature Extractors.** The feature extractors for different models are summarized in Table 2. All networks use the ReLU activation function.

Table 2: Network architectures for different models.

| Model | Fully Connected Layers | GRU Hidden Size | GRU Output Size |
|---|---|---|---|
| PG-Kmeans | (128, 128) | 64 | 128 |
| VAE/DEC/CAAE-Encoder | (128, 128) | 64 | 128 |
| VAE/DEC/CAAE-Decoder | (128, 32, 32) | N/A | N/A |
| VAE/DEC/CAAE-Encoder-Attention-Heads | | 2 | |
| VAE/DEC/CAAE-Encoder-Attention-Feature-Size | | 2×8 | |

| Task | Single-run PG-Kmeans | PG-Kmeans | DEC | DEC (best of 5)* |
|------|---------------------|-----------|-----|------------------|
| halfcheetah | 0.495 (50% $\in$ [0.99, 1.00]) | **0.989** $\pm$ 0.000 | 0.945 $\pm$ 0.031 | **0.989** $\pm$ 0.000 |
| ant | 0.745 (85% $\in$ [0.83, 0.94]) | **0.924** $\pm$ 0.003 | 0.390 $\pm$ 0.512 | 0.756 $\pm$ 0.003 |
| walker2d | 0.557 (50% $\in$ [0.70, 0.99]) | **0.942** $\pm$ 0.005 | 0.767 (70% $\in$ [0.99, 1.00]) | **0.990** $\pm$ 0.000 |
| hopper | 0.258 (25% $\in$ [0.99, 1.00]) | **0.994** (80% $\in$ [0.99, 1.00]) | 0.000 $\pm$ 0.000 | 0.000 $\pm$ 0.000 |
| Diagonal | 0.892 $\pm$ 0.032 | **0.920** $\pm$ 0.001 | 0.276 $\pm$ 0.017 | 0.287 $\pm$ 0.001 |
| Takeball | **0.996** $\pm$ 0.002 | **0.997** $\pm$ 0.000 | 0.054 $\pm$ 0.097 | 0.175 $\pm$ 0.027 |

Table 3: Normalized Mutual Information (NMI) scores for all methods on D4RL and GridWorld environments, averaged over at least 10 random seeds. For experiments with clearly distinguishable success or failure outcomes, we report the probability of success along with the NMI range conditioned on successful trials. For all other cases, we assume Gaussian noise and report the 95% confidence interval. *Note: DEC (best of 5) is not a practically feasible algorithm, as ground truth labels are unavailable in real-world clustering scenarios. Without access to true labels, selecting the best result is infeasible. In contrast, PG-Kmeans (best of N) remains a valid approach, as it relies on the final objective function $J(W, \theta)$ as an internal metric to determine the best clustering result for output.

| Task | Single-run PG-Kmeans | PG-Kmeans | BC | 10%-BC | AWAC | CQL |
|------|---------------------|-----------|-----|--------|------|-----|
| halfcheetah | 56.5 | 83.4 | 55.9 | 90.1 | 93.6 | 95.6 |
| ant | 76.3 | 130.4 | / | / | / | / |
| walker2d | 44.3 | 104.5 | 99.0 | 108.7 | 49.4 | 109.6 |
| hopper | 42.48 | 87.2 | 52.3 | 111.2 | 52.7 | 99.3 |

Table 4: Normalized returns of PG-Kmeans and baseline offline RL algorithms on D4RL datasets. Results for other methods are taken from the CORL benchmark Tarasov et al. (2022). PG-Kmeans is initialized with four cluster centers, and the reported returns are obtained by evaluating the center policies of the two non-trivial (non-zero) clusters in each environment. The results indicate that PG-Kmeans significantly outperforms Behavior Cloning. However, for efficiency reasons, center policies are not fully trained to convergence during optimization. As a result, even with nearly perfect clustering, the learned center policies may not always serve as optimal action generators. Notably, PG-Kmeans operates as a semi-supervised learning method, requiring only a minimal amount of return signals for evaluation, yet achieving performance comparable to fully supervised RL algorithms.

## C  APPENDIX: FULL RESULTS AND ABLATION STUDIES

### C.1  FULL RESULTS

Here, we provide experimental results for single-run PG-Kmeans and Best-of-5 DEC for comparison with PG-Kmeans. Notably, even when DEC is allowed to enhance its performance by repeatedly running and selecting the best result, it still fails to achieve satisfactory classification in certain environments. This limitation arises because DEC lacks an explicit policy clustering representation, meaning that its embedding-based clustering approach does not guarantee successful categorization in complex environments. Also, as we discussed in Section 6.2, there is no significant negative correlation between loss and NMI for CAAE algorithm, so we didn't do Best-of-5 experiment for CAAE algorithm.

We also evaluated the performance of PG-Kmeans' centroid strategy, the experimental results are presented in Table 8, with classification histograms shown in Figure 6. Across all four environments, PG-Kmeans successfully classified trajectories generated by different policies. When $k > 5$, the accuracy of classification exceeded 50% in all cases. In most cases, the best output policy reaches the performance of agents trained with 10% BC(Tarasov et al., 2022).

### C.2  ABLATION STUDIES

#### C.2.1  REGULARIZATION IN CAAE

To mitigate the overfitting issue, we introduced a regularization term to the codebook in CAAE. Without this regularization, for the codebook $\mu$, the last layer of the encoder $C$ and the first layer of

Table 5: NMI of CAAE with different cluster counts $k$ in different environments. The results are averaged over 10 random seeds.

| Task | k=4 | k=5 | k=6 | k=7 | k=8 |
|---|---|---|---|---|---|
| Ant | **0.88** | 0.87 | 0.83 | 0.81 | 0.79 |
| HalfCheetah | **0.90** | 0.85 | 0.81 | 0.80 | 0.75 |
| Hopper | **0.66** | 0.55 | 0.50 | 0.47 | 0.46 |
| Walker2d | **0.79** | 0.74 | 0.71 | 0.69 | 0.64 |
| Diagonal | N/A | 0.82 | **0.87** | 0.80 | 0.83 |
| Takeball | **1.00** | 0.95 | 0.90 | 0.88 | 0.85 |
| PathFollowing | 0.18 | 0.22 | 0.27 | 0.34 | **0.35** |
| Extra | 0.38 | 0.34 | **0.43** | 0.36 | 0.37 |

Table 6: Running time of PG-Kmeans and CAAE. The unit of time is minute in this table. The GPU used is NVIDIA RTX A6000, 48GB.

| Algorithm | HalfCheetah | Ant | Walker2d | Hopper | Diagonal | Takeball | PathFollowing | Extra |
|---|---|---|---|---|---|---|---|---|
| PG-Kmean | 17 | 24 | 32 | 70 | 22 | 5.7 | 24 | 9.7 |
| CAAE | 6.6 | 10 | 9.7 | 15 | 2.4 | 1.9 | 38 | 2.3 |

Table 7: NMI of CAAE with or with out codebook regularization.

| Regularization | HalfCheetah | Ant | Walker2d | Hopper | Diagonal | Takeball | PathFollowing | Extra |
|---|---|---|---|---|---|---|---|---|
| W/o | 0.99 | 0.96 | 0.65 | 0.99 | 0.84 | 1.00 | 0.14 | 0.39 |
| W/ | 0.99 | 0.96 | 0.88 | 0.99 | 0.86 | 1.00 | 0.15 | 0.43 |

the decoder $D$, and for a real number $0 < \lambda < 1$. We can find the recon loss will not change when $C' = \lambda C, D' = \lambda D$, but the codebook loss will being smaller if $\mu' = \lambda \mu$. This phenomenon will lead the codebook to collapse to a single point, which is not desirable. To prevent this, we add a regularization term to the codebook loss.

We can see from Table 7, regularized CAAE is more robust and achieves better performance in some hard environments. And regularization will not hurt the performance in almost all the environments.

### C.2.2 IMPACT OF INITIAL CLUSTER COUNT $k$.

We further examined the impact of the initial cluster count $k$ on the clustering performance to assess the algorithm's robustness when the true number of categories $k^*$ is unknown or inaccurately estimated. Due to poor performance in the GridWorld environment, this analysis focuses solely on the Gym dataset.

As shown in Figure 5, DEC is highly sensitive to $k$. On the Gym dataset, increasing $k$ to 10 results in a significant performance drop ($0.8 \rightarrow 0.5$). This degradation primarily stems from the emergence of multiple active cluster centers, which fragment a single true class—an inherent limitation of K-means-based methods. PG-Kmeans exhibits a similar issue but to a lesser extent. On the Gym dataset, it does not naturally disperse, and in the more challenging GridWorld dataset, its NMI decreases by less than 0.1 even when $k$ is increased to three times the ground-truth value. Furthermore, after applying the merging process, PG-Kmeans experiences almost no performance degradation. In fact, when $k$ is slightly larger than $k^*$, it benefits from reduced overfitting, leading to improved performance.

As illustrated in Table 5, CAAE's clustering performance does not exhibit consistent patterns like PG-Kmeans or DEC, but rather demonstrates distinct and divergent trends across different datasets – showing a clear positive correlation in Pathfollowing environments, a negative correlation in Walker2d, and only minor fluctuations likely caused by variance in Extra. Generally, more challenging environments benefit more from increasing the k-value, while simpler ones show the opposite trend. We believe that this phenomenon may stem from the aforementioned limitations of the K-means method, which not only compromises model performance but also mitigates the overfitting issue discussed earlier.

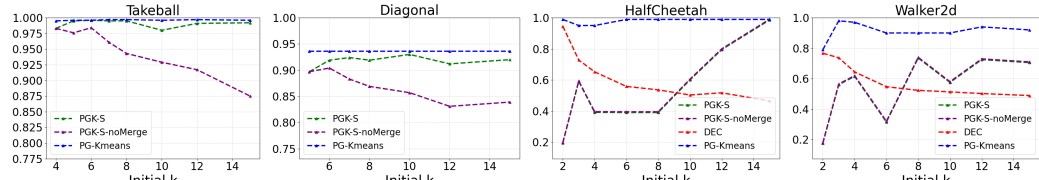

Figure 5: The impact of different initial cluster counts $k$ on the final NMI. All reported values are averaged over 10 random seeds. PGK-S represents the results of single-run PG-Kmeans, while PGK-S-noMerge represents single-run PG-Kmeans without merging. As shown in the figure, increasing the initial cluster count $k$ leads to a noticeable decline in performance metrics for both PG-Kmeans and DEC, with the effect being more pronounced in DEC. This is because a single class is more likely to be split into multiple subclasses that cannot be automatically merged. However, after introducing the merging process, this issue no longer significantly affects PG-Kmeans' performance. Additionally, a higher initial $k$ reduces the probability of cluster fusion, thereby stabilizing PG-Kmeans training. Consequently, in the HalfCheetah and Walker2d environments, increasing the initial $k$ actually improves PG-Kmeans' clustering performance.

Another advantage of increasing $k$ is the enhanced stability of PG-Kmeans, as it reduces the likelihood of mode collapse. Across all tested Gym environments, we observe that the probability of correctly identifying the two policies increases as $k$ grows.

Empirically, for PG-Kmeans algorithm, the optimal choice of $k$ should be slightly larger than $k^*$, as this strikes a balance between improving stability, reducing overfitting, and minimizing classification accuracy loss.

Table 8: Highest normalized evaluation returns of $k$ output policies, where the expert performance is scaled to 100.0. Every policy is evaluated on 10 different random seeds. Because we stopped the training once all the trajectories have got clear preference over some certain policy, so some networks were not fully trained for evaluation and those results are marked with *. We also highlighted all the results that clearly suffered from mode collapse in red. In these experiments, almost all trajectories were assigned to the same cluster.

| Env \ $k$ | 3 | 4 | 5 | 6 | 7 | 8 | 10 | 12 | 15 |
|---|---|---|---|---|---|---|---|---|---|
| HalfCheetah | 40.47 | 42.13 | 45.75* | 32.58* | 69.79 | 68.47 | 28.76 | 88.95 | 49.59* |
| Hopper | 44.42 | 45.56 | 46.47 | 111.24 | 80.84* | 69.85* | 93.44 | 24.07* | 101.33 |
| Ant | 22.27 | 116.91 | 111.94 | 61.97 | 103.52 | 130.80 | 47.78* | 67.38 | 67.80 |
| Walker2d | 107.44 | 8.69* | 109.84 | 62.74 | 103.32 | 1.29* | 98.26 | 6.33* | -0.09* |

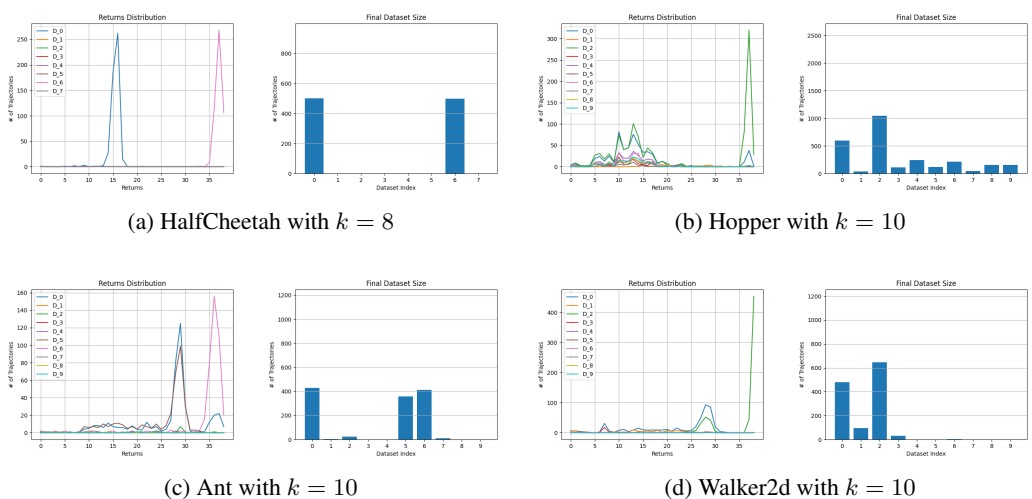

(a) HalfCheetah with $k = 8$        (b) Hopper with $k = 10$

(c) Ant with $k = 10$        (d) Walker2d with $k = 10$

Figure 6: Successful classification samples for different environments and $k$ values. From the figures, it is evident that the dataset is divided into 2–3 primary components. Since the dataset does not come with inherent classification labels, it is not possible to directly calculate classification accuracy. However, we can infer the differences between clusters indirectly by analyzing the return distribution of each cluster.

# D   ADDITIONAL DISCUSSION: CLUSTERING FOR DOWNSTREAM RL TRAINING

To further investigate the utility of trajectory clustering for downstream policy learning, we conducted experiments on the four D4RL medium-expert datasets (Ant, HalfCheetah, Hopper, and Walker2d). Since the GridWorld dataset does not provide reward signals, it was excluded from this study.

We applied CAAE to cluster trajectories, then sampled 10 trajectories from each cluster to estimate expected returns. CQL and IQL were subsequently trained only on the cluster with the highest average return, denoted as "–clustered." Each result is averaged over 9 random seeds. For PG-Kmeans ($k = 5$), we sampled 10 trajectories per cluster to estimate cluster returns and directly evaluated the centroid policy of the highest-return cluster without additional RL training. Implementations of CQL and IQL followed the JAX-CORL framework.[1]

Table 9: Performance of CQL, IQL, and PG-Kmeans with and without dataset clustering on D4RL medium-expert tasks.

| Method + Dataset | CQL-All | CQL-Clustered | IQL-All | IQL-Clustered | PG-Kmeans |
|---|---|---|---|---|---|
| Ant | 127.11 | 93.72 | 122.55 | 123.14 | 121.94 |
| HalfCheetah | 38.00 | 48.23 | 91.83 | 93.67 | 69.79 |
| Hopper | 1.64 | 76.98 | 35.40 | 105.51 | 97.33 |
| Walker2d | 106.46 | 79.98 | 108.37 | 108.71 | 107.44 |

The results indicate that clustering does not universally outperform training on the full dataset, but it can yield substantial improvements in specific problem–algorithm combinations, particularly in Hopper and HalfCheetah with IQL. Two additional observations are worth noting:

- **Minimal reliance on reward signals.** PG-Kmeans achieves competitive performance without any additional RL training, relying only on return samples for cluster selection and centroid evaluation.
- **Clustering as a practical tool.** Although clustering cannot guarantee improvements in every downstream setting, it provides a flexible mechanism to reorganize heterogeneous datasets

---

[1]https://github.com/nissymori/JAX-CORL

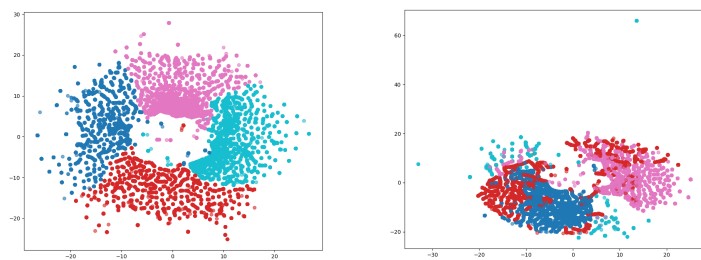

Figure 7: t-SNE visualization of the latent spaces on the TakeBall environment. CAAE (left) produces compact and semantically coherent clusters, while VQ-VAE (right) exhibits fragmented and less stable clusters.

and selectively exploit high-quality behaviors, thereby enhancing the practical applicability of offline RL.

# E  ADDITIONAL DISCUSSION: CAAE VS. VQ-VAE

In the main text, we reported that VQ-VAE constitutes a strong baseline, already outperforming DEC and in many cases surpassing SORL. However, CAAE consistently achieves superior clustering stability and accuracy. Here we provide a more detailed comparison.

The key differences lie in the latent representation design:

- **Continuous vs. discrete representation.** CAAE directly learns continuous latent vectors, avoiding the discrete codebook projection used in VQ-VAE. This eliminates quantization errors during training.
- **Stability during early training.** VQ-VAE relies on the Straight-Through Estimator, which is known to cause instability in the early stages due to gradient mismatch. CAAE circumvents this issue by optimizing in continuous space.
- **Codebook competition.** In VQ-VAE, multiple codebook vectors may compete for assignment, leading to unstable optimization. CAAE avoids this bottleneck and achieves faster encoder–decoder convergence.
- **Balanced gradient flow.** VQ-VAE sometimes suffers from gradient monopolization by a small subset of codebook entries, which biases representation learning. CAAE distributes gradients more evenly across latent dimensions, leading to more balanced policy-semantic learning.

We further illustrate this comparison in Figure 7, which visualizes clustering on the TakeBall environment via t-SNE. The VQ-VAE representation exhibits noticeable fragmentation, whereas CAAE produces more compact and semantically coherent clusters.

These results confirm that while VQ-VAE is a competitive baseline, the design of CAAE offers distinct advantages for stable and interpretable trajectory clustering.

