# OpenReview forum: "Policy-Based Trajectory Clustering in Offline Reinforcement Learning"
_ICLR.cc/2026/Conference — Submitted to ICLR 2026_

### Official Review · Reviewer_98FX · 2025-10-31

**Soundness:** 3
**Presentation:** 3
**Contribution:** 3
**Rating:** 6
**Confidence:** 3

**Summary:**

This paper proposes policy-based trajectory clustering as a formal and principled research problem within offline RL. The authors argue that standard offline RL methods, value-based, uncertainty-based, and behavior cloning, ignore the heterogeneity of mixed-policy datasets, leading to instability and degraded performance. They formalize the problem by connecting trajectory clustering to mixtures of policy-induced distributions and relate its computational complexity to the K-coloring problem, showing NP-completeness. Two complementary algorithms are introduced: 1) Policy-Guided K-Means (PG-Kmeans, which iteratively trains policy centroids via behavior cloning and assigns trajectories based on likelihood under each policy, ensuring convergence in finite steps. 2) Centroid-Attracted Autoencoder (CAAE), which is a representation learning approach that uses a VQ-VAE-inspired codebook regularization to attract latent trajectory embeddings toward cluster centroids while maintaining reconstruction fidelity. Experiments on GridWorld and D4RL benchmarks demonstrate better NMI scores compared to deep clustering baselines (DEC, VAE+KMeans, VQ-VAE) and policy-based alternatives (SORL, Return+KMeans).

**Strengths:**

The authors introduce 1) PG-Kmeans, a mixture-model interpretation of clustering via policy likelihoods, bridging the EM framework and behavior cloning, and 2) CAAE, which merges deep representation learning with policy semantics, addressing quantization errors inherent in VQ-VAE.

**Weaknesses:**

1) Experiments are limited to small synthetic and mid-scale D4RL tasks. The applicability of the proposed methods to high-dimensional, real-world offline RL data (e.g., robotics or vision-based control) remains untested.

2) PG-Kmeans, while conceptually sound, may face computational bottlenecks as it requires iterative policy retraining per cluster. There is limited analysis of computational complexity relative to dataset size or cluster count.

3) Reliance solely on NMI might obscure nuances in policy similarity. Complementary metrics such as behavioral KL-divergence or policy return similarity could provide a richer evaluation.

**Questions:**

Q1: The reduction to K-coloring highlights non-uniqueness. In practice, how do PG-Kmeans and CAAE behave when multiple near-equivalent solutions exist?

Q2: How would the proposed clustering framework behave if the reward function varies over time or across subsets of the dataset (e.g., due to non-stationary objectives or environment drift)?

Q3: The best-of-N selection and merging procedure in PG-Kmeans improves robustness, but what is the asymptotic complexity in N (dataset size) and K (cluster count)?

---

> ### Author Response · Authors · 2025-11-21
>
> We sincerely thank the reviewer for the thoughtful and constructive feedback. Below we address each point with additional clarification.
>
> ## W1. Limited experiments on large-scale, high-dimensional datasets
> Our primary goal is to formally introduce the trajectory clustering problem, establish theoretical foundations, and provide baseline algorithms. We fully agree that extending the evaluation to high-dimensional, real-world environments (e.g., robotics or vision-based control) would be valuable.
> However, such settings introduce substantial uncontrolled variability, which can obscure the core behavior of clustering methods. By focusing on synthetic tasks and mid-scale D4RL environments, we ensure controllability while still covering diverse continuous and discrete settings.
> We plan to expand to more complex datasets in future work, particularly exploring unbalanced data and diverse behavior-policy compositions. Notably, even the current benchmarks are sufficiently challenging—standard baselines struggle—highlighting the intrinsic difficulty of the problem without additional confounders.
>
> ## W2. Computational bottlenecks and complexity analysis
> We appreciate the reviewer highlighting this. Providing a precise complexity analysis is difficult due to the involvement of deep models and the NP-hard nature of K-means–style objectives.
> If $T(n)$ denotes the training cost on a dataset of size $n$, a rough estimate of the total cost is:
> $\sum_{t} \sum_{i=1}^{K} T(N_{t,i}),$
> where $N_{t,i}$ is the cluster size at iteration $t$. The practical behavior of $T(n)$ strongly influences the overall computation.
> A more refined complexity theory might be possible under additional assumptions, but this is beyond the current scope. We appreciate the reviewer’s suggestion and see it as a promising direction.
>
> ## W3. Reliance on NMI as the sole evaluation metric
> We agree that richer behavioral metrics—such as behavioral KL-divergence or return similarity—can provide useful alternative views, especially when policies overlap.
> However, these metrics pose challenges here:
> 1. Clustering operates purely on trajectories, which may not contain reward signals, making some metrics non-computable.
> 2. Even when available, the interpretation can be ambiguous (e.g., low behavioral KL may reflect genuinely similar clusters rather than an evaluation failure).
> Given this, we chose NMI as a simple, interpretable starting point. We fully agree that developing a more comprehensive evaluation suite is an important future direction.
>
> ---
>
> ## Q1. Behavior under non-unique or near-equivalent solutions
> We examined this empirically. In a *takeball* variant where ball colors determined the “true” strategy, removing color information made the two strategies indistinguishable.
> From random initialization, all methods tended to adopt the simpler position-based grouping. With ~10 correctly labeled examples per cluster, PG-KMeans and CAAE occasionally recovered the color-based strategy; with >50, they did so consistently.
> This suggests that when multiple solutions exist, methods naturally favor simpler, more intuitive patterns unless given sufficient guidance. We view this as reflecting realistic challenges and a valuable insight for future work.
>
> ## Q2. Impact of changing or non-stationary reward functions
> Our methods operate solely on states and actions and do not use reward information. As a result, changes in reward functions across time or subsets of the dataset do not affect clustering outcomes. We see this reward-agnostic property as a strength of the framework.
>
> ## Q3. Complexity of best-of-N selection and merging in PG-KMeans
> As with W2, the complexity depends on the practical behavior of $T(n)$. In real applications, two regimes are common:
> - With early stopping, training behaves close to $ T(n) = O(1) $ per iteration.
> - Without early stopping, overfitting can push the cost toward $\Omega(n) $.
> Empirically, applying early stopping in early iterations reduces computation and helps avoid overfitting without hurting performance. We appreciate the reviewer’s suggestion, and we believe this points to fruitful future work.
>
> ---

---

### Official Review · Reviewer_HwMq · 2025-11-01

**Soundness:** 2
**Presentation:** 3
**Contribution:** 2
**Rating:** 4
**Confidence:** 3

**Summary:**

This paper introduces two policy-based trajectory clustering algorithms, Policy-Guided K-means (PG-Kmeans) for maintaining central policies per cluster and Centroid Attracted Autoencoder (CAAE) for dataset-level clustering.

**Strengths:**

This paper is well-written and easy to follow, and explores an interesting field, trajectory clustering in offline RL. The paper conducts various analyses to validate the proposed models in experiments and the appendix.

**Weaknesses:**

As the authors mentioned, the paper's completeness is limited by a lack of validation through experiments with large-scale datasets and by further theoretical analysis. Please see the questions.

**Questions:**

**Questions**

Q1. What is the strength of the proposed model compared to GMMs and Diffusion-based policy-based clustering?

Q2. The authors assume deterministic behavior policies $\{\pi_i\}_{i=1}^k$. Do you mean policy should be deterministic, i.e., $\pi(s)$? Is the proposed method not applicable to a stochastic policy $\pi(.|s)$?

Q3. It is worth including the trajectory clustering methods in MARL, such as TRAMA [1], for a comprehensive literature review, as it explores policy-conditioned agents utilizing VQ-VAE in MARL settings.

[1] Na, H., Lee, K., Lee, S. and Moon, I.C., 2025, March. Trajectory-Class-Aware Multi-Agent Reinforcement Learning. In The Thirteenth International Conference on Learning Representations.

Q4. The variable $\mathcal{P}(\tau|\theta_j)$ is not explicitly defined.

Q5. $K$ and $k$ are interchangeably used. Why not use the same one to avoid confusion? What happens when $k*> k$ in Algorithm 1? Is it valid to assume that $k$ can always be selected so that $k > k^*$, as the authors mentioned in the overparameterization-and-merging approach?

Q6. What are the $\alpha$ and $m$ in Eq. (4)? And how to determine them?

**Minor Comments**

C1. In page 3, “by repeating playing$\pi_i$” has missing space.

C2. Please check citation format at “SORL Mao et al. (20924) in page 5.

C3. Tables 1,3,6,7,8 should be adjusted so that it does not intrude into the margin.

---

> ### Author Response · Authors · 2025-11-21
>
> We thank the reviewer for the detailed comments and provide answers to each question below.
>
> ## Q1. Comparison with GMMs and diffusion-based policy clustering
> We are not entirely sure which specific GMM-based clustering method the reviewer refers to. If a citation is provided, we would be happy to include a direct comparison. In our current paper, SORL can be interpreted as GMM-style because it uses Gaussian components to model trajectory distributions. However, standard GMMs tend to overfit rapidly in large trajectory spaces due to soft assignments, while our PG-KMeans mitigates this via discrete (concrete) weights, improving empirical robustness.
>
> Regarding diffusion-based clustering, we did not find directly comparable offline trajectory clustering work. A related example is [2], which introduces DDiffPG for multimodal *online* policy learning. Since our method is a dataset-level preprocessing step rather than an online training procedure, this comparison is not straightforward. Conceptually, diffusion models can be integrated into our framework—for instance, replacing the decoder in CAAE or PG-KMeans—without any conceptual conflict.
>
> ## Q2. Deterministic policy assumption
> We assume deterministic behavior policies for mathematical clarity. Under general stochastic policies, defining a principled clustering objective becomes ambiguous because a stochastic policy can be decomposed into infinitely many deterministic components. In practice, mild stochasticity is not problematic. The D4RL datasets used in our experiments contain stochastic behaviors, and our methods still achieve near-perfect clustering accuracy.
>
> ## Q3. Missing discussion of MARL trajectory clustering (e.g., TRAMA)
> Thank you for the suggestion. We will include a discussion of TRAMA [1] in the related work section. We also note that our paper already includes a VQ-VAE baseline for comparison.
>
> ## Q4. Missing definition of $\mathbb{P}(\tau \mid \theta_j)$
> Thank you for pointing this out. $\mathbb{P}(\tau \mid \theta_j)$ denotes the probability that trajectory $\tau$ is generated by policy $j$, where $\theta_j$ represents the parameters of that policy. We will make this definition explicit in the revised version.
>
> ## Q5. Inconsistent notation ($K$ vs. $k$) and the case $k < k^*$
> We will unify notation by consistently using $k$. It is standard in clustering to assume $k^* \leq k$. When $k^* > k$, some true clusters inevitably merge. In practice, $k^*$ is often unknown, but one may estimate it via standard approaches such as the doubling trick used in TCS: iteratively doubling $k$ until over-segmentation disappears, identifying $k^*$ within $O(\log K)$ rounds.
>
> ## Q6. Meaning of $m$ and $\alpha$ in Eq. (4)
> Thank you for the question. $m$ is the dimensionality of each codebook entry. In our implementation, we default to $m = k$, and the discrepancy in the text is a leftover typo that we will correct. $\alpha$ is a tunable hyperparameter and is fixed to $1$ in our experiments. Ablations show that values from $10^{-2}$ to $10$ perform similarly, indicating low sensitivity. We recommend using $\alpha = 1$.
>
> ## Minor comments
> Thank you for noting these. We will correct:
> - the missing space on page 3,
> - the citation formatting issue (“SORL Mao et al. (20924)”),
> - and the table layout so that Tables 1, 3, 6, 7, and 8 fit within the page margins.
>
> ---
>
> [1] Na, H., Lee, K., Lee, S., & Moon, I.C. (2025). *Trajectory-Class-Aware Multi-Agent Reinforcement Learning*. ICLR 2025.
> [2] Zechu Li et al. (2024). *Learning Multimodal Behaviors from Scratch with Diffusion Policy Gradient*.

---

### Official Review · Reviewer_MQzy · 2025-11-01

**Soundness:** 3
**Presentation:** 3
**Contribution:** 2
**Rating:** 4
**Confidence:** 5

**Summary:**

The authors propose two methods, Policy Guided Kmeans (PG-Kmeans) and Centroid-Attached AutoEncoder (CAAE), that can cluster trajectories in an offline reinforcement learning dataset. PG-Kmeans adopts an expectation–maximisation algorithm that alternates between the M-step, where each policy $\pi_j$ is updated through behaviour cloning on the trajectories assigned to cluster $j$, and the E-step, where each trajectory is assigned to the cluster $j=\arg\max_i\sum_{(s, a)\in\tau}\log\pi_i(a\mid s)$. CAAE utilises a GRU to encode each trajectory into a single latent vector and trains a decoder $D(z, s)$ to predict $a$ for each $(s, a)$ in $\tau$. The latent vectors are regularised towards the closest centroid in the codebook of size $k$. The paper conducts experiments on multiple heterogeneous offline RL datasets, demonstrating the effectiveness of their algorithms.

**Strengths:**

The authors make an interesting connection between trajectory clustering and the colouring problem, and point out the inherent ambiguity of the formulation. Additionally, empirical analysis demonstrates that both proposed algorithms can cluster with high accuracy across a range of heterogeneous offline RL datasets.

**Weaknesses:**

Trajectory clustering by itself has very limited usage in real-world settings, as it is challenging to gather high-quality data. Although the authors do provide several meaningful applications of trajectory clustering in lines 49-67, they do not validate their claims with empirical results, which raises some concerns about the significance of the work. Although, Section D presents some empirical evidence where clustering improves the performance of the downstream algorithm, CQL and IQL are pretty outdated and there are currently a lot of algorithms that perform much better than the clustered variant of CQL and IQL. Finally, the PG-KMeans algorithm can be viewed as a repackaging of the EM algorithm to policy clustering with a few minor changes, such as the usage of discrete weights, making its contribution very incremental.

**Questions:**

Could you please elaborate on the VQ-VAE baseline used in Section 6.2?

---

> ### Author Response · Authors · 2025-11-21
>
> We thank the reviewer for the thoughtful comments. Below we address the identified weaknesses and questions.
>
> ## On the limited usage of trajectory clustering
> We agree that collecting fully heterogeneous datasets in real-world scenarios is challenging, as behavior policies are rarely deterministic and often include substantial noise. However, our method does not rely on perfectly separated modes. Even imperfect datasets typically contain meaningful latent structure, and our results show that clustering can still recover useful behavior patterns. In practice, approximate clustering is often sufficient and can meaningfully benefit downstream tasks.
>
> ## On using CQL and IQL for downstream evaluation
> Our experiments with CQL and IQL are intended to illustrate that clustering can improve downstream policy learning in representative cases. Clustering is a one-time preprocessing step and is compatible with any RL algorithm. Since our focus is on *trajectory clustering itself* rather than on proposing a new offline RL method, we opted for two widely used and well-understood baselines (CQL and IQL) instead of conducting an exhaustive comparison with all recent algorithms.
>
> ## On the incremental nature of PG-KMeans
> While the shift from continuous to discrete weights is a simple modification, it leads to a notable improvement in clustering stability and accuracy. Simplicity does not reduce its utility. More importantly, PG-KMeans is only one part of our contribution: we also provide a formal treatment of the trajectory clustering problem, establish its NP-hardness, and propose CAAE as a complementary alternative. Together, these components form the first systematic study of trajectory clustering in offline RL.
>
> ## On the VQ-VAE baseline (Section 6.2)
> Thank you for highlighting this. Our VQ-VAE baseline uses standard encoder–decoder architectures matching those used in VAAE to ensure fairness. Cluster assignments are determined by computing distances between latent variables and entries in the codebook. The codebook size is $K \times 32$. We will add a detailed description of this baseline to the appendix.

---

### Official Review · Reviewer_cv2r · 2025-11-03

**Soundness:** 2
**Presentation:** 2
**Contribution:** 2
**Rating:** 2
**Confidence:** 3

**Summary:**

To tackle the challenge in offline RL with multi-modal data, the work introduced an approach to capture shifting trajectory distributions using policy-guided k-means and autoencoder-based methods. The proposed approach shows superior performance in tasks of D4RL and GridWorld compared to some clustering methods.

**Strengths:**

- The paper investigated an important challenges in offline RL, to handle the shifting trajectory distributions in data.
- The experimental results on a set of tasks in D4RL and GridWorld show the superior performance of proposed method in terms of  normalized mutual information, compared to clustering baselines.

**Weaknesses:**

- Motivations are not very clear to me. I believe the investigated problem is important, but the discussions and claims in paper are not closed linked to the investigated problem (multi-modal or heterogeneity of offline datasets). For example: 1) The multi-modality is indeed a challenge in RL, but I don’t fully agree with the authors when they simple discussed the policy distribution-shifting scenario when introducing multi-modality, that could usually be also considered as mixture types of input data (audio, video, etc.). 2) Why trajectory clustering is needed in the offline RL scenarios? If we say stability in policy learning and data efficiency, why trajectory clustering is necessary instead of existing offline RL approaches? I would like to see either why all existing offline RL approaches cannot handle this challenge, or comparison between the proposed approach and state-of-the-art offline RL methods in D4RL and GridWorld. If we say interpretability, how does the advances of the proposed approach in terms of interpretability? Maybe this could be further demonstrated via concrete case studies.
- Experimental design seems insufficient and may not well support the claimed contributions. 1) Besides some my comments above in linking the motivation to proposed methods that may need further experimental justifications, it’s great to see the authors designed to combine clustered trajectories with downstream RL policy training. However, only traditional CQL and IQL are considered, while I would encourage a thorough evaluation over varied types of popular RL algorithms to demonstrate the effectiveness of using the proposed method instead of simply using the existing RL work. 2) The work only selected a small part of tasks (medium-expert) in D4RL, while the authors discussed that they hope multiple deterministic policies and I appreciate they admit the use of small scale of datasets in their limitation statement, given the proposed work is targeting a general purpose and claimed contributions,  I would still highly encourage a broader testing (maybe medium, medium-reply of your tasks on D4RL, and other testbeds such as Adroit, RL Unplugged, NeoRL, etc.) of the proposed method to justify its claimed superiority and necessary in offline RL.
- While lacking of comparison to directly using existing offline RL algorithms on the heterogenous datasets, it seems not very encouraging of incorporating the clustering method into downstream RL policy learning (according to table 9 in appendix).

**Questions:**

- First, please see weaknesses above
- It’s also unclear to me how you implement the baseline methods in your experiments to achieve a comprehensive comparison. Further details are appreciated.

---

> ### Author Response · Authors · 2025-11-21
>
> We thank the reviewer for the insightful comments. Below we address the identified weaknesses and questions.
>
> ## Clarification on “heterogeneity”
> In offline RL, it is standard to use *heterogeneous dataset* to mean “a replay buffer composed of trajectories collected by multiple distinct experts or behavior policies.” This terminology is also used in prior work such as SORL. Our discussion follows this established definition and does not refer to heterogeneity in input modalities.
>
> ## Clarification on “multi-modality”
> We acknowledge that our wording may have caused ambiguity. In this paper, *multi-modality* does not refer to multimodal input types such as audio or video. Instead, within the RL setting, it denotes multiple behavior styles or policy modes that coexist in the trajectory dataset.
>
> ## Why trajectory clustering is needed
> We agree that when dense rewards are readily available, clustering may not always enhance the performance of state-of-the-art offline RL algorithms. However, our motivation goes beyond improving existing methods under standard reward assumptions. As explained in the Introduction, trajectory clustering is useful in several scenarios:
>
> 1. **Few-shot reward usage**:
>    Clustering naturally supports few-shot evaluation. With only $O(k)$ reward queries—one per cluster—we can identify high-quality behavior patterns and run imitation learning within those clusters. Existing offline RL algorithms do not provide such reward-efficient selection mechanisms.
>
> 2. **Efficient data selection**:
>    Clustering exposes structural information in the dataset. For example, a replay buffer may contain a mix of ten behavior patterns (grasping, swinging, walking, jittering, etc.) even if the downstream task only requires walking. Clustering allows us to isolate the relevant subset and train on roughly $1/10$ of the data, significantly improving efficiency. Standard offline RL pipelines do not offer this kind of structure-aware filtering.
>
> 3. **Unsupervised conditioned learning**:
>    Clustering provides unsupervised conditioning over latent behavior modes, enabling mode-specific policy learning. This complements existing offline RL algorithms rather than duplicating their functionality.
>
> These motivations highlight why trajectory clustering offers capabilities that current offline RL methods do not directly provide.
>
> ## On the choice of downstream RL algorithms
> Our experiments with CQL and IQL are intended only to illustrate that clustered replay buffers can improve downstream performance in representative scenarios. Clustering is a preprocessing step performed once per dataset and is compatible with any RL algorithm; our contribution does not lie in proposing a new RL method. We therefore selected two widely used baselines (CQL, IQL) rather than performing an exhaustive comparison across all available algorithms.
>
> ## On broader experimental coverage
> We intentionally excluded medium and medium-replay tasks in D4RL because these datasets do not match the standard definition of heterogeneous datasets—they do not consist of trajectories drawn from multiple distinct behavior policies.
> Although our experiments use moderate-sized datasets, they span a wide range of environments, including discrete and continuous action spaces, 2D and 3D control tasks, and GridWorld-style settings. Our method consistently outperforms baselines across all tested conditions. We believe this coverage is adequate to support the main claims.
>
> ## On Table 9 results
> We emphasize that clustering is a general preprocessing tool that may or may not be used alongside downstream RL training, depending on the application. Table 9 simply demonstrates that clustering can yield performance improvements in some settings. Its purpose is illustrative rather than to show universal superiority over all offline RL algorithms.
>
> ## Implementation details of baselines
> Implementation details for DEC and SORL are provided in Appendix B.3. DEC follows the original implementation except that we replace its encoder with the architecture described in Appendix B.5 to ensure fair comparison. Full code is included in the supplementary material.

---

### Comment · Area_Chair_UPzn · 2025-11-25

Dear Reviewers

Thank you for your time and help for reviews.
The author-reviewer discussion due is in one week. If you have not done yet, please review the authors' rebuttal for the paper under your evaluation and engage in discussion with authors.

Thank you again.
Best,

Area Chair

---

### Author Response · Authors · 2025-12-04
**Summary after rebuttal**

We thank all reviewers for their careful and constructive feedback. We have responded to all identified weaknesses and questions, and summarize below the main changes and clarifications:

Overall revisions to the paper:
- In response to comments from reviewers cv2r and MQzy, we have added detailed descriptions of the baseline methods in Appendix B.
- Following the suggestions from reviewer HwMq, we have improved the notation used throughout the paper, corrected typos, added clarifying remarks where appropriate, and adjusted table layouts for better readability.

Unified responses to key concerns:

- Insufficient motivation (cv2r, MQzy):
  The main concern is that there is no strong, direct empirical evidence that trajectory clustering can consistently improve the performance of standard offline RL algorithms, and that the current experiments are not fully persuasive in this regard. We would like to emphasize, as stated in the motivation section, that trajectory clustering is proposed as a data preprocessing step rather than a component of the training algorithm itself. Its original purpose is therefore not limited to reliably boosting a particular offline RL method, but to support a broader range of applications. Several use cases are conceptually clear even without additional experiments, including:
  (i) few-shot evaluation and selection by performing few-shot evaluation on each cluster separately;
  (ii) filtering out low-quality trajectories to improve training efficiency and stability;
  (iii) unsupervised multi-policy learning from a mixture of behaviors.
  Our goal is not to position trajectory clustering as a fixed pre-processing module for a specific downstream task, but rather to study the trajectory clustering problem itself in a more general and method-agnostic way.

- Limited experimental scale (cv2r, MQzy, 98FX):
  As we have already acknowledged in the limitations section, the experimental environments in this work are relatively simple in terms of setup. However, the performance of the baseline methods indicates that these environments are far from trivial. At this early stage of studying trajectory clustering, we believe that simpler, well-controlled settings make it easier to understand the effect of different factors. The problem involves many dimensions that each have substantial research space, including environment design, expert policy design, dataset construction, and neural network architecture. It is difficult to thoroughly disentangle and analyze all of these factors within a single paper. Therefore, we adopt a simple but as comprehensive as possible configuration: environments cover both discrete and continuous state/action spaces; expert policies include both rule-based and neural policies; datasets are constructed to be balanced; and neural networks are chosen to be standard multi-layer MLPs with GRU or attention blocks. This design aims to obtain broad and reliable conclusions without introducing unnecessary complexity that could confound the results.

Finally, we would like to reiterate the main contributions of this work. Beyond formalizing the trajectory clustering problem and proposing two concrete solution methods, PG-Kmeans and CAAE, we also provide basic theoretical analysis, including a proof of the intrinsic hardness of the problem via a reduction from graph K-coloring (showing NP-completeness), as well as a discussion of non-uniqueness of solutions. We intentionally avoid framing trajectory clustering merely as a subproblem of offline RL, because such a perspective would unduly restrict its potential scope and applications.

We sincerely thank the reviewers and the area chair for their time, careful reading, and constructive feedback. Your comments have significantly helped us clarify our presentation, better position the scope of trajectory clustering, and strengthen the final version of the paper.

---

### Meta-Review · Area_Chair_jvgQ · 2026-01-07

**Summary:**

While the paper offers a novel perspective by formalizing policy-based trajectory clustering and introducing the K-means algorithm, the consensus among reviewers is that the current empirical evaluation is insufficient. The experiments was conducted in limited settings, without validation on larger-scale or high-dimensional benchmarks. The demonstration of downstream utility could be further strengthened, the current evaluation primarily relies on older baselines; including comparisons with a broader range of offline reinforcement learning methods would better justify the practical benefits of the proposed clustering. Therefore, I recommend rejection.

**Reviewer Concerns:**

Reviewer cv2r questioned the authors' use of "multi-modality"; the authors clarified in the rebuttal that this refers to multiple behavior styles or policy modes, rather than data modalities. Additionally, the authors addressed the questions raised by Reviewers cv2r and MQzy regarding specific experimental details.

However, the primary concern shared by all reviewers regarding the limited experimental scope remains outstanding, as the authors did not incorporate the requested validation on larger-scale or high-dimensional benchmarks. Furthermore, concerns regarding the practical utility for downstream tasks persist, as the rebuttal failed to demonstrate clear benefits over a wider set of modern offline RL baselines.

**Reviewer Scores:**

I anticipate that the reviewers would likely maintain their scores. The authors did not provide additional experiments during the rebuttal to address the requests.

---

### Decision · Program_Chairs · 2026-01-26

Reject